# Tissue Distributions and Toxic Effects of Hexavalent Chromium in Laboratory-Exposed Periwinkle (*Littorina littorea* Linnaeus)

**DOI:** 10.3390/ani13213412

**Published:** 2023-11-03

**Authors:** Olufemi S. Salami, Joseph A. Adeyemi, Toluwase S. Olawuyi, Fernando Barbosa, Chris O. Adedire

**Affiliations:** 1Department of Biology, School of Life Sciences, Federal University of Technology, Akure P.O. Box 704, Nigeria; ossalami@futa.edu.ng (O.S.S.); coadedire@futa.edu.ng (C.O.A.); 2Department of Clinical Analyses, Toxicology and Food Sciences, School of Pharmaceutical Sciences of Ribeirão Preto, University of São Paulo, Avenida do Café s/no, Ribeirão Preto 14040-903, Brazil; fbarbosa@fcfrp.usp.br; 3Department of Human Anatomy, School of Basic Medical Sciences, Federal University of Technology, Akure P.O. Box 704, Nigeria; tsolawuyi@futa.edu.ng

**Keywords:** hexavalent chromium, condition index, tissue distribution, proximate analyses, periwinkles, histopathology

## Abstract

**Simple Summary:**

The increased application of chromium compounds in several fields has resulted in elevated levels of toxic hexavalent chromium (Cr^6+^) in the aquatic environment, thus creating the potential for bioaccumulation of Cr^6+^ in the tissues as well as eliciting various toxic effects in organisms. The present study investigated the effects of Cr^6+^ exposure on the tissue distribution, proximate composition, and histopathology of an aquatic mollusk, periwinkle (*Littorina littorea*). The animals were exposed to sublethal concentrations of Cr^6+^ (0.42, 0.84, and 4.2 mg/L) for 30 days. Exposure to Cr^6+^ resulted in changes in the proximate composition and histological architecture of *L. littorea*. There was a low potential for bioaccumulation of Cr^6+^ in the tissue of *L. littorea*. Thus, its consumption did not pose any serious health risks to humans.

**Abstract:**

The increased use of hexavalent chromium (Cr^6+^) in various industrial applications has contributed to its elevated levels in the environment, especially the aquatic environment. Thus, there is the potential for accumulation of Cr^6+^ in the tissues of aquatic organisms and consequent toxic effects. The toxic effects of Cr^6+^ in aquatic organisms have been widely reported; however, little is known about the patterns of tissue accumulation of Cr^6+^ and its toxicity in aquatic mollusks. Thus, the present study investigated the effects of Cr^6+^ exposure on the tissue distribution, proximate composition, and histopathology of an aquatic mollusk, periwinkle (*Littorina littorea*). The animals were exposed to sublethal concentrations of Cr^6+^ (0.42, 0.84, and 4.2 mg/L) for 30 days, after which the condition index, tissue accumulation, proximate composition, and histopathological effects were determined. The control animals were maintained in a medium that did not contain Cr^6+^ (0 mg/L). The condition index did not differ significantly among the groups. The levels of Cr^6+^ in the tissues differed significantly among the different tissue types while there was no significant effect of the exposure concentration, except in the foot tissue. The proximate parameters (protein, carbohydrates, lipid, crude fiber, and moisture contents) differed significantly among the groups. The protein contents of the exposed animals were significantly lower than those of the control animals and the histological architecture of the major organs was altered in the chromium-exposed animals. The findings from this study indicate a low potential of *L. littorea* to bioaccumulate Cr^6+^ in its tissues at the low exposure concentrations tested in this study; as such, its consumption may not pose any serious health risks to humans. However, changes in the proximate composition and histological architecture of the exposed *L. littorea* show that Cr^6+^ is potentially toxic to periwinkles.

## 1. Introduction

Hexavalent chromium (Cr^6+^) is one of the stable oxidation states of the transition metal chromium, and the adjudged most toxic of all its oxidation states [1,2]. Hexavalent chromium has been listed as a chemical of public health concern [3,4,5]. Similarly, it has been rated as a group 1 carcinogen by the International Agency for Research on Cancer (IARC) [6,7]. The environmental levels of hexavalent chromium compounds have increased tremendously in recent years due to their industrial applications in the production of stainless steel, leather tanning, wood preservation, electroplating, etc. [8]. The United States annual production of sodium dichromate in the year 2015 was put at 280 million pounds [9]. Hence, hexavalent chromium has been detected in industrial effluents in different parts of the world at levels that were above the permissible limit of 20 µg/L in drinking water [10,11,12]. Specifically, a concentration as high as 7.65 mg/L of chromium was detected at a site that was close to a tannery in Uttar Pradesh, India [13].

The aquatic ecosystems mostly serve as the repository for the discharged environmental contaminants, which predisposes resident organisms to various physiological disturbances and ultimately death [14,15]. The toxic effects of hexavalent chromium on aquatic organisms have been reported; the exposure of Indian major carp (*Catla catla*) to a sublethal concentration of hexavalent chromium resulted in increased frequencies of micro- and binucleated erythrocytes, as well as increased DNA in the DNA tail [16]. Similarly, Nagpure et al. [17] showed that the exposure of a freshwater fish, *Labeo rohita* to sublethal concentrations of hexavalent chromium resulted in an increased percentage tail DNA in the erythrocytes and gills of the exposed fish compared to the control fish, and thus concluded that hexavalent chromium was genotoxic to the fish. The acute exposure of tadpoles of *Lithobates catesbeianus* to 4, 12 or 36 mg/L hexavalent chromium resulted in significantly higher frequencies of micronuclei and induction of histopathological lesions such as necrosis, hepatocytes with pyknotic nuclei, loss of cellular integrity, inflammatory infiltrates, congested sinusoids, and the presence of melanomacrophages in the liver and inflammatory infiltrates, granulomas, tubular and glomerular hypertrophy, and tubular necrosis in the kidney of the exposed tadpoles [18]. Fernando et al. [19] reported that the chronic exposure of the tadpoles and adults of the Asian common toad, *Duttaphrynus melanostictus* to sublethal concentrations of hexavalent chromium (0.002–2.0 mg/L) altered its hematological parameters, and caused a significant increase in the frequencies of erythrocytic micronuclei and the percentage tail DNA. In the Olive Flounder, *Paralichthys olivaceus*, exposure to 1.0 or 2.0 mg/L waterborne hexavalent chromium caused increased activities of enzymes involved in redox homeostasis as well as immune dysfunction evidenced in reduced activity of lysozyme and immunoglobulin M levels [20]. Also, the exposure of the bivalve, *Venus verrucosa* to 1, 10, or 100 µg/L hexavalent chromium for 7 days resulted in oxidative stress and altered the fatty acids composition [21].

Periwinkles are gastropods that are commonly found in the intertidal zones, and are an important source of dietary proteins for humans and other animals [22,23]. A number of studies have shown that periwinkles have the potential to bioaccumulate toxicants, especially metals in their tissues [24,25], and since metals react strongly with biological systems, this may result in damage to body cells and tissues. The details of tissue accumulation patterns of hexavalent chromium and the damage to tissues in periwinkle are not fully understood. Furthermore, there are reports that the nutritive values of edible aquatic organisms are compromised due to exposure to toxicants [26,27]. However, there is a paucity of information on the effects of exposure to hexavalent chromium on the proximate composition of periwinkles. It is against this background that this study aimed to investigate the effects of exposure to hexavalent chromium on the proximate compositions of periwinkles. The tissue distribution of hexavalent chromium, and the histopathological analyses of major tissues of the periwinkle, *L. littorea* were performed.

## 2. Materials and Methods

### 2.1. Experimental Organism

The periwinkles (Shell length: 4–5 cm; average weight: 4.2 ± 0.8 g) used in this study were obtained from brackish water in Abealala, Ilaje Local Government, Ondo State, Nigeria (6°21′18″ N 4°66′10″ E). The environmental salinity at the point of the collection was determined in situ and was found to be 7.9 ppt. Water samples were collected at the collection point, and both the periwinkles and water samples were transported in clean plastic containers to the Postgraduate Research Laboratory, Department of Biology of the Federal University of Technology, Akure, Nigeria. The organisms were allowed to acclimatize to the laboratory environment for fourteen (14) days prior to commencement of experiments, and fed with water leaf, *Talinum triangulare*. The background concentration of total chromium in water samples collected from the collection point was found to be 0.3 mg/L using atomic absorption spectrophotometry.

### 2.2. Laboratory Exposure of Periwinkles to Hexavalent Chromium

Firstly, the laboratory-acclimatized animals were divided into seven groups and transferred into plastic containers (150 × 80 × 50 mm) and were exposed to 0, 3.5, 7.0. 10.5, 14.0, 17.5, or 21.0 mg/L of Cr^6+^ for 96 h. The choice of the exposure concentrations was based on previous range-finding experiments performed in the laboratory. The test solutions were prepared from a 1000 mg/L stock solution of hexavalent chromium, which was previously prepared from a potassium dichromate salt. The plastic containers were acid-washed prior to use for experiments. Each container contained twenty experimental organisms, and there were three replicates of each treatment. The animals were monitored for survival during the exposure duration at 24 h intervals. An animal is adjudged dead when completely retracted into its shell or failed to protrude the foot during 15 min of observation. The survival data were subjected to Probit analysis using the Statistics Package for Social Sciences (SPSS), version 21 for the determination of median lethal concentration (LC_50_).

Thereafter, in another set of experiments, the animals were exposed for thirty (30) days to three sublethal concentrations; 0.42, 0.84, and 4.2 mg/L being ^1^/_20_, ^1^/_10_, and ½ of the 96 h LC_50_ of Cr^6+^, respectively. The animals were exposed to sublethal concentrations to enhance survival during the exposure duration. The exposure medium was changed every four days during the experiment to prevent the accumulation of excretory products in the experimental medium. The animals’ exposure to Cr^6+^ was performed in plastic containers (150 × 80 × 50 mm), with twenty animals per container, and there were three replicates of each treatment. The control animals were maintained in water not contaminated with Cr^6+^. The containers were covered with muslin cloth to allow for access to oxygen but prevented the animals from escaping during experiments. At the end of the exposure, representative samples of the animals were crushed, and the soft tissues were excised, and frozen at −20 °C for later determination of proximate composition and metal accumulation in the tissues. A portion of each tissue was fixed in formaldehyde for histopathological analysis.

### 2.3. Determination of Levels of Chromium in Periwinkle Tissues

The level of chromium was determined following the procedures described by Adeyemi and Deaton [28], with minor modifications. The excised soft tissues (head, gill, mantle, foot, and intestine) were rinsed with deionized water and dried in an oven at 60 °C for 48 h in order to obtain a constant weight. The dried tissues were subsequently digested in a mixture of nitric acid and hydrochloric acid (ratio 1:3 *v*/*v*), and the digests were made up to volume with distilled water. The total chromium in the tissue digests was determined using a flame atomic absorption spectrophotometer (Buck 235). The calibration curve was obtained by plotting the absorbance values against the concentrations of working standard solutions (0.5–10 mg/L), which were prepared from a 1000 mg/L stock solution. The absorbance value of the reagent blank was determined ten times, and the method’s limit of detection was determined using the formula:LOD = 3 SD/m
where LOD is the limit of detection, SD is the standard deviation of 10 blank determinations and m is the slope of the calibration curve. The LOD was found to be 0.092 ± 0.06 mg/L.

### 2.4. Determination of Condition Index (CI)

At the end of the 30-day exposure, 20 animals were sub-sampled from each experimental group, and the total wet weights of the soft tissue and the whole organism were determined using a weighing balance (JA3003 model). The condition index (CI) was determined following the method of Michel et al. [29] as follows:CI = (wet weight of soft tissue/wet weight of the whole organism) × 100

### 2.5. Determination of Proximate Compositions of Periwinkle Tissues

The proximate compositions of the periwinkles were determined according to the methods of the Association of Official Analytical Chemists [30]. The proximate parameters determined were moisture, ash, carbohydrate, crude fiber, and crude protein. The ash content was determined by drying the tissues in an oven at 550 °C for 20 h until a white residue was obtained. The moisture content in the fish tissue was estimated by drying the samples in an oven, set at 105 °C for 8 h until a constant weight was obtained. The moisture content is the difference in weight between the wet weight and the dry weight. The protein content was determined by digesting the tissue with sulfuric acid inside a micro-Kjeldahl flask. The tissue digests were then diluted with sodium hydroxide and distilled water. Boric acid was used to collect the released ammonia, and the total nitrogen was determined titrimetrically. The crude fiber content was determined by digesting the sample in sulfuric acid for under reflux. Thereafter, the digests were filtered and washed with hot water. The residues were boiled in potassium hydroxide, filtered, and washed in a mixture of water and acetone, and the residues were dried in the oven at 105 °C, so as to obtain a constant weight. The lipid content was determined by homogenizing about 2 g of the tissues in a chloroform–methanol mixture (2:1 *v*/*v*) and then centrifuged at 1000× *g* for 10 min. The upper layer was gently removed with a Pasteur pipette while the lower portion was allowed to evaporate to dryness, and the weight of the residue was taken. The carbohydrate content was estimated by finding the summation of all the proximate parameters determined and subtracting the sum from 100 g.

### 2.6. Histopathological Analyses of the Soft Tissue

The procedures described by Adeyemi et al. [31], with slight modifications, were followed for the histopathological analyses. The soft tissues (brain, gills, intestine, foot, and mantle) were fixed in 10% formaldehyde for 24 h, after which they were dehydrated in ethanol gradients. The tissues were then cleared with xylene, and embedded in paraffin wax. Sections of about 7 µm were cut onto clean slides from the embedded tissues using a microtome (AM-202A manual rotary). The slides were then stained with hematoxylin/eosin dyes, and observed under a microscope at ×40 objective, and the photomicrographs of the desired views were taken.

### 2.7. Data Analysis

Data were presented as the mean ± standard deviation. The survival data obtained from the acute toxicity experiment were subjected to Probit analysis for the determination of median lethal concentration (LC_50_). The differences in the means of tissue accumulation of hexavalent chromium were tested using a two-way analysis of variance (the factors being the tissue types and exposure concentration) while the differences in the means of data from other parameters were determined using a one-way analysis of variance, and the means were separated using New Duncan’s Multiple Range Test. Statistical significance was assumed at *p* < 0.05. Statistical analyses were performed using the Statistics Package for Social Sciences (SPSS), version 21.

## 3. Results

### 3.1. Mortality Rate

The percentage mortality of the periwinkles exposed to different concentrations of Cr^6+^ is shown in Table 1. The percentage mortality was both concentration- and exposure-duration-dependent. The percentage mortality increased significantly with increasing exposure concentration, at the four exposure time points (*p* < 0.05). Also, the percentage mortality was higher at longer exposure durations, especially at 72 and 96 h exposure durations.

### 3.2. Condition Index (CI)

The condition index of *L. littorea* exposed to sublethal concentrations of Cr^6+^ is presented in Figure 1. The condition index did not differ significantly (*p* > 0.05) among the groups.

### 3.3. Tissue Accumulation of Total Chromium in L. littorea Exposed to Sublethal Concentrations

Table 2 shows the levels of total chromium in the brain, gills, mantle, foot, and intestine of *L. littorea* exposed to sublethal concentrations of Cr^6+^. The exposure concentration did not have a significant effect on the accumulation of total chromium in the tissues of *L. littorea* except in the foot where the level of chromium increased significantly at the highest concentration (4.2 mg/L). For all the exposure concentrations and the control, the total levels of chromium differed significantly among the tissues (*p* < 0.05). Specifically, the levels of total chromium were highest in the brain followed by the intestine, and least in the foot. The total chromium tissue accumulation ranges were 0.67–0.78, 0.30–0.48, 0.40–0.48, 0.57–0.69, and 0.46–0.58 µg/g in the brain, foot, gills, intestine, and mantle, respectively.

### 3.4. Proximate Compositions of the Tissues

The proximate compositions of the tissues of *L. littorea* exposed to sublethal concentrations of hexavalent chromium are presented in Table 3. Overall, the protein contents in the tissues decreased significantly with increasing exposure concentration except in the mantle where the protein levels were statistically similar. However, the carbohydrate content increased with increasing exposure concentration except in the mantle where the carbohydrate levels decreased significantly in the group that was exposed to the highest hexavalent chromium concentration (4.2 mg/L). In the brain, the protein and crude fiber contents decreased significantly while the carbohydrate and moisture contents increased significantly in the exposed groups compared to the control (*p* < 0.05). However, the crude lipid levels were statistically similar for all the exposure groups, except for the group exposed to 0.42 mg/L hexavalent chromium in which the lipid levels were significantly higher. The range values of protein, carbohydrate, lipid, fiber, and moisture were 19.38–29.89, 4.74–9.50, 1.08–2.63, 3.55–6.05, and 58–64%, respectively.

In the foot, the levels of protein decreased significantly in the animals that were exposed to hexavalent chromium while the levels of carbohydrate and crude fiber increased significantly (*p* < 0.05). There was no significant difference in the moisture contents among the groups (*p* > 0.05). Additionally, the lipid contents were statistically the same for most of the groups except in the animals that were exposed to 0.42 mg/L in which there was a significant increase in the lipid contents. The range values of protein, carbohydrate, lipid, fiber, and moisture were 19.18–22.68, 11.72–14.07, 1.56–2.90, 2.42–5.86, and 59–61.66%, respectively.

In the gills, the protein contents decreased significantly while the carbohydrate and lipid levels increased significantly in the exposed groups, especially in the animals exposed to 4.2 mg/L (*p* < 0.05). The crude fiber levels were significantly higher in the animals exposed to 0.42 and 0.8 mg/L hexavalent chromium compared to the control. The moisture content was highest in the animals exposed to 4.2 mg/L hexavalent chromium compared to the other groups. The range values of protein, carbohydrate, lipids, crude fiber, and moisture contents were 15.36–32.15, 2.42–9.36, 1.96–9.85, 2.39–3.80, and 59–67%, respectively.

In the intestine, the percentage of protein, carbohydrate, fiber, and lipids increased significantly in the animals that were exposed to hexavalent chromium especially those exposed to 0.84 and 4.2 mg/L. The moisture content was significantly higher in the animals exposed to the highest concentration compared to the other groups (*p* < 0.05). The range values of protein, carbohydrate, lipids, crude fiber, and moisture contents were 10.61–28.12, 6.72–10.70, 1.88–7.39, 3.10–8.30, and 59–63%, respectively.

In the mantle, the protein and crude fiber content did not differ significantly among the groups (*p* > 0.05). There was a significant decrease in the carbohydrate and moisture contents in the exposed animals, especially at the highest exposure concentration while the lipid contents were significantly higher in the exposed animals. The range values of protein, carbohydrate, lipids, crude, and moisture contents were 17.66–18.86, 8.30–11.51, 4.81–10.31, 2.51–3.53, and 59–63%, respectively.

### 3.5. Histopathology of the Soft Tissues of L. littorea

#### 3.5.1. Brain

Figure 2A–D shows the transverse sections of the brain of *L*. *littorea* chronically exposed to 0, 0.42, 0.84, and 4.2 mg/L Cr^6+^, respectively. Figure 2A shows the normal histology of the brain of *L. littorea* in which there was no atrophy, and the white and grey matter of the neural cells were clearly shown. The animals exposed to 0.42 mg/L Cr^6+^ had visible lesions in the brain (Figure 2B). The exposure of *L. littorea* to 0.84 mg/L resulted in the pia matter becoming widened, as well as the atrophy of neurons at the boundary of grey and white matter (Figure 2C). The animals that were exposed to 4.2 mg/L Cr^6+^ showed a high degree of deterioration of neural cells and lacked a clear distinction between the grey and white matter (Figure 2D).

#### 3.5.2. Gills

The histological sections of the gills of *L. littorea* exposed to Cr^6+^ are shown in Figure 3. The histological sections of the control animals showed the normal histoarchitecture of the gills of *L. littorea* in which the pillar cells, mucous cells, pavement cells, and erythrocytes are all clearly identified (Figure 3A). The animals that were exposed to 0.42 mg/L Cr^6+^ showed necrosis and atrophy of secondary lamella of the pavement and pillar cells (Figure 3B). There was atrophy of the secondary lamella in animals exposed to 0.84 and 4.20 mg/L Cr^6+^ (Figure 3C and Figure 3D, respectively).

#### 3.5.3. Mantle

Figure 4 shows the transverse section of the mantle of *L*. *littorea* chronically exposed to Cr^6+^. There was no observable lesion in the section of the mantle of control animals (Figure 4A) and those exposed 0.42 mg/L Cr^6+^ (B), in which the mantle epithelium (shown in Figure 4A,B) and secretory cells (Figure 4A alone) were visible. There was degeneration of secretory cells and mantle epithelium in the exposed animals (Figure 4C,D).

#### 3.5.4. Intestine

The histological sections of the intestine of *L. littorea* exposed to Cr^6+^ are shown in Figure 5. The histological section of the control animals did not reveal any observable lesion. The Crypts of Lieberkuhn, ectasia of the intestinal lumen, mucosa, muscularis mucosae, villi, and submucosa were visible (Figure 5A). The animals that were exposed to 0.42 mg/L Cr^6+^ showed necrosis of the basal cells (Figure 5B). There was a degradation of basal cells and infiltration of hemocytes into the villi in the sections of animals exposed to 0.84 mg/L Cr^6+^ (Figure 5C). The exposure of *L. littorea* to 4.2 mg/L Cr^6+^ resulted in the widening of the lumen as well as ectasia of the villi around the lumen. The epithelial cells of the villi that enclosed the lumen showed some degree of degradation (Figure 5D).

#### 3.5.5. Foot

Figure 6 shows the transverse sections of the foot of *L*. *littorea* chronically exposed to Cr^6+^. The control animals showed normal histoarchitecture of the foot in which the hyaline, muscle fibers (myofibers), and epithelium were visible (Figure 6A). The exposed animals showed the atrophy of myofibers (Figure 6B), swelling of the epithelium, and degeneration of myofibers (Figure 6C,D).

## 4. Discussion

The results from this study indicate a significant difference in the levels of chromium in the different tissues of the periwinkle, *L. littorea*. A number of studies have reported the accumulation of toxic hexavalent chromium in the different tissues of aquatic animals [32,33]. The accumulation of metals in the tissues of animals has been shown to depend on factors such as exposure concentration, exposure duration, species, and tissue type [33,34,35]. In the present study, chromium accumulation was highest in the brain, intestine, and mantle of *L. littorea.* These results were consistent with findings from similar studies that determined the tissue accumulation of metals in aquatic mollusks. Gundacker [36] reported high concentrations of metals such as cadmium, lead, copper, and zinc in the viscera (internal organs including the intestine) of freshwater mollusks. Also, Vernon et al. [37] reported the highest accumulation of metals including chromium in the digestive tract of the marine bivalve, *Mytilus galloprovincialis*. There have been disparities in the levels and patterns of tissue-specific accumulation of hexavalent chromium in laboratory-exposed aquatic organisms. For example, the gills and digestive glands levels of hexavalent chromium in the bivalve mollusk, *Venus verrucosa* were 180 and 150 µg/g, respectively, while a tissue concentration as low as 1.73 µg/g in the gills of a fish chronically exposed to 5.3 mg/L hexavalent chromium [21,31]. Comparatively, the levels of hexavalent chromium in the tissues of periwinkles were low ranging from 0.40 to 0.48 µg/g in the gills to 0.67–0.78 µg/g in the brain. This low accumulation of chromium in the soft tissues of *L. littorea* indicates that the dietary route through consumption of periwinkles does not pose a serious health risk to human health.

The condition index has been routinely used as an indicator of exposure of animals to environmental pollutants [38,39]. The condition index did not differ significantly among the exposure groups in this study. The non-significant difference in the condition index in the exposed animals has also been reported in a study by Neves et al. [38] in which periwinkles were exposed to various concentrations of copper. This may be due to the metabolizing activities of some metal-detoxifying enzymes and proteins such as cytochrome P450 and metallothioneins, which can bind to cytosolic metals, thus reducing their adverse effects in the exposed organisms [40,41,42]. Generally, the condition index decreases in response to exposure to environmental pollutants [38,43], and this may be due to the toxicants interfering with the normal life activities of the animals.

The proximate composition of an animal reveals its nutritive value. Periwinkles are consumed as a delicacy in several parts of the world for their richness in proteins and omega-3 fatty acids [44]. Chronic exposure to hexavalent chromium significantly altered the proximate composition of the animals, especially the protein content, which reduced significantly in all the tissues of the exposed animals, especially at the highest exposure concentration. The decreased protein content in the exposed animals has been reported in other studies [45,46]. The protein content of the fish African catfish decreased significantly following exposure to chlorpyrifos [45]. Also, Sohail et al. [46] reported a significant decrease in the protein contents in the tissues of the freshwater mussel, *Anodonta anatina.* The decreased protein content could be due to interference of the metal with the process of protein synthesis in the exposed animals [47].

The changes in the histological architecture have been used as indices of the physiological effects of pollutants on organisms [30,31]. The tissue-specific histological changes in animals have been shown to relate to the extent of pollutant accumulation in the tissue [48,49]. In the present study, exposure to hexavalent chromium resulted in several changes in the histological structures of the tissues of *L. littorea* such as atrophy of neuronal cells, necrosis and atrophy of the gills’ secondary lamellae, degeneration of mantle’s secretory cells, degradation of the villi epithelial cells, and atrophy and degeneration of foot’s myofibers. Although the reports of histopathological effects of hexavalent chromium in periwinkle and aquatic mollusks are scarce, a number of studies have shown that exposure to chromium resulted in histological changes in the tissues of fish and other aquatic animals [18,31,50,51,52]. The histopathological changes are physiological responses to defend the animals against assaults by toxicants [31]. For example, histological changes such as hyperplasia and hypertrophy, and fusion of lamellae have been identified as defense mechanisms to minimize the toxic effects of pollutants in fish [53].

## 5. Conclusions

The levels of hexavalent chromium differed significantly among the tissue types but this was not due to the exposure concentration since for the levels of hexavalent chromium were statistically similar between the control and the exposed groups for most of the tissues. Exposure to hexavalent chromium caused a significant change in the proximate compositions of the animals. In particular, the protein contents of the exposed animals were significantly lower than those of the control animals. The histological architecture of the major organs was altered in the chromium-exposed animals. Overall, the findings from this study indicate a low potential of *L. littorea* to bioaccumulate Cr^6+^ in its tissues; as such, its consumption did not pose any serious health risks to humans. However, changes in the proximate composition and histological architecture of the exposed *L. littorea* show that Cr^6+^ is potentially toxic to periwinkles.

## Figures and Tables

**Figure 1 animals-13-03412-f001:**
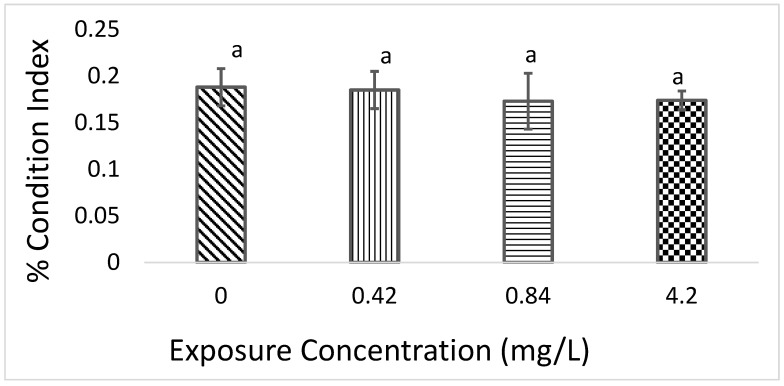
The condition index of *L. littorea* exposed to sublethal concentrations of Cr^6+^. Each bar is the mean ± standard deviation of three replicates. Bars represented with the same alphabets are not significantly different.

**Figure 2 animals-13-03412-f002:**
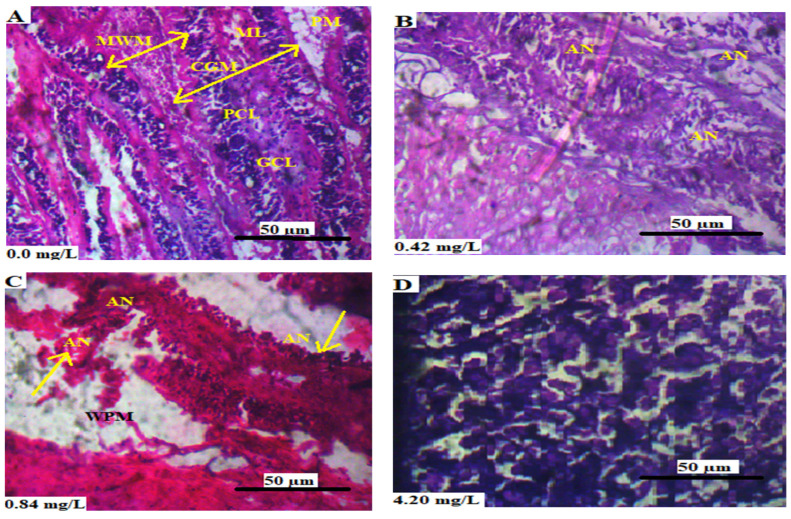
(**A**–**D**) are the transverse sections of the brain of *L. littorea* cronically exposed to 0, 0.42, 0.84, and 4.2 mg/L Cr^6+^, respectively. Showing MWM: medulla of white matter, ML: molecular layer, CGM: cortex of grey matter, PM: pia mater, PCL: Purkinje cell layer, GCL: granule cell layer, AN: atrophy of neurons and WPM: widened pia mater. The tissues were stained with H&E and viewed at magnification × 40.

**Figure 3 animals-13-03412-f003:**
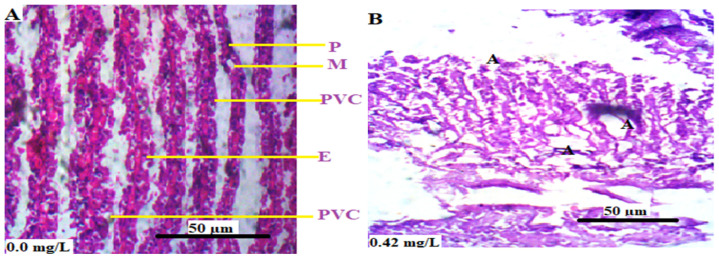
(**A**–**D**) are the transverse sections of the gills of *L. littorea* chronically exposed to 0, 0.42, 0.84, and 4.2 mg/L Cr^6+^, respectively. Showing P: pillar cells, M: mucous cell, PVC: pavement cells, E: erythrocyte, A: area of atrophy and N: area of necrosis. The arrows in (**D**) show the secondary lamellae. The tissues were stained with H&E and viewed at magnification × 400.

**Figure 4 animals-13-03412-f004:**
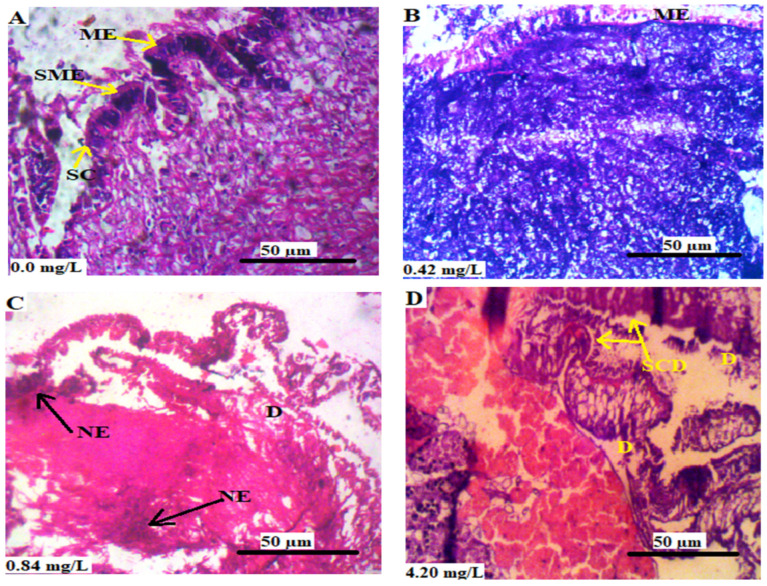
(**A**–**D**) are the transverse sections of the mantle of *L. littorea* chronically exposed to 0, 0.42, 0.84, and 4.2 mg/L Cr^6+^, respectively, for 30 days. Showing SME: secretory mantle epithelium, ME: mantle epithelium, SC: secretory cell, SCD: secretory cells degenerated, NE: necrosis of epithelium and D: area of degeneration. The tissues were stained with H&E and viewed at magnification × 400.

**Figure 5 animals-13-03412-f005:**
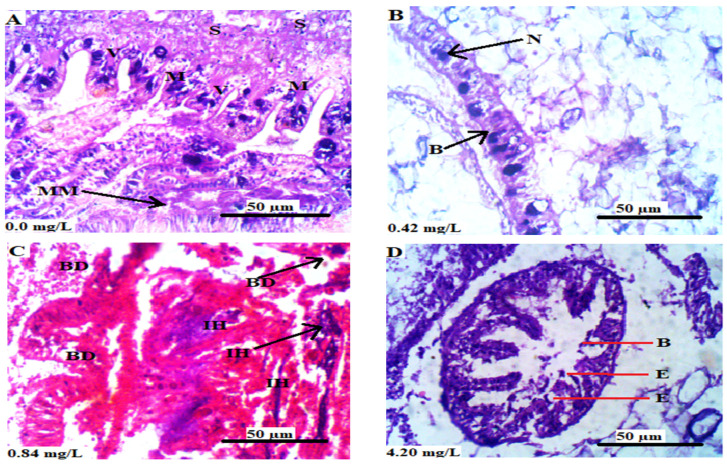
(**A**–**D**) are the transverse sections of the intestinal tissues of *L. littorea* chronically exposed to 0, 0.42, 0.84, and 4.2 mg/L Cr^6+^, respectively, for 30 days. Showing B: basal cell, BD: degradation of basal cells CL: Crypts of LieberkÜhn E: ectasia of villi around the intestinal lumen, M: mucosa, MM: muscularis mucosae, V: villi, S: submucosa, IH: infiltration of hemocytes, and N: area of necrosis. The tissues were stained with H&E and viewed at magnification × 400.

**Figure 6 animals-13-03412-f006:**
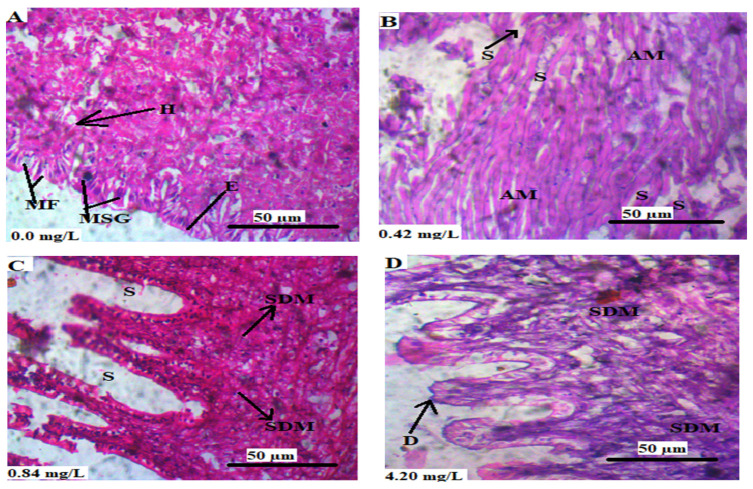
(**A**–**D**) are the transverse sections of the foot tissues of *L. littorea* chronically exposed to 0, 0.42, 0.84, and 4.2 mg/L Cr^6+^, respectively, for 30 days. Showing H: hyaline, MF: muscle fiber, MSG: mucous-secreting gland, E: epithelium, S: space/gap, AM: atrophy of myofiber, SDM: swelling and degeneration of myofibers, and D: degenerative area. The tissues were stained with H&E and viewed at magnification × 400.

**Table 1 animals-13-03412-t001:** Percentage mortality of *L. littorea* exposed to hexavalent chromium (Cr^6+^).

Concentration of Cr^6+^ (mg/L)	Percentage Mortality (Inactive) of *L. littorea* per Day			
	24 h	48 h	72 h	96 h
Control	0.00 ± 0.0 ^a^	0.00 ± 0.0 ^a^	0.00 ± 0.00 ^a^	0.00 ± 0.00 ^a^
3.5	15.00 ± 5.7 ^b^	15.00 ± 5.8 ^b^	52.50 ± 5.0 ^b^	55.00 ± 5.8 ^b^
7.0	22.50 ± 5.0 ^b,c^	35.00 ± 17.3 ^c^	52.50 ± 5.0 ^b^	65.00± 5.8 ^c^
10.5	32.50 ± 9.5 ^c,d^	45.00 ± 17.3 ^c^	70.00 ± 11.5 ^c^	75.00± 5.8 ^d^
14.0	35.00 ± 5.8 ^c,d^	67.50 ± 5.00 ^d^	75.00 ± 5.8 ^c^	85.00 ± 5.8 ^e^
17.5	35.00 ± 5.8 ^c,d^	85.00 ± 5.8 ^e^	85.00 ± 5.8 ^d^	85.00 ± 5.0 ^e^
21.0	45.00 ± 17.6 ^d^	95.00 ± 5.8 ^e^	95.00 ± 5.8 ^e^	95.00 ± 5.8 ^f^

Each value is the mean ± standard deviation of four replicates. Values followed by the same superscript letters in the same column are not significantly different.

**Table 2 animals-13-03412-t002:** Total chromium (µg/g) in the different tissues of *L. littorea* sublethal concentrations of Cr^6+^.

Exposure Concentration (mg/L)	Tissue Accumulation (µg/g)				
	Brain	Foot	Gills	Intestine	Mantle
Control	0.67 ± 0.03 ^a,C^	0.30 ± 0.03 ^a,A^	0.40 ± 0.08 ^a,A,B^	0.57 ± 0.05 ^a,B^	0.46 ± 0.05 ^a,A,B^
0.42	0.78 ± 0.05 ^a,C^	0.45 ± 0.06 ^a,b,A,B^	0.41 ± 0.09 ^a,A^	0.64 ± 0.05 ^a,B,C^	0.58 ± 0.05 ^a,A,B^
0.8	0.74 ± 0.02 ^a,C^	0.39 ± 0.05 ^a,b,A^	0.48 ± 0.14 ^a,A,B^	0.66 ± 0.04 ^a,B,C^	0.57 ± 0.02 ^a,A,B,C^
4.2	0.70 ± 0.06 ^a,B^	0.48 ± 0.05 ^b,A^	0.46 ± 0.05 ^a,A^	0.69 ± 0.06 ^a,B^	0.54 ± 0.06 ^a,A,B^

Each value is the mean ± standard deviation of three replicates; values followed by the same upper case letters in the same row are not significantly different while values followed by the same lower case letters in the same column are not significantly different.

**Table 3 animals-13-03412-t003:** Effects of hexavalent chromium on proximate composition of the tissues of *L. littorea*.

Body Part	Conc. (mg/L)	Protein (%)	Carbohydrate (%)	Lipid (%)	Fiber (%)	Moisture (%)
Brain	Control	29.89 ± 1.69 ^c^	4.74 ± 0.86 ^a^	1.32 ± 0.02 ^a,b^	6.05 ± 0.18 ^c^	58.00 ± 1.00 ^a^
	0.42	22.63 ± 0.32 ^b^	7.19 ± 0.23 ^b^	2.63 ± 0.29 ^c^	3.55 ± 0.10 ^a^	64.00 ± 0.00 ^b^
	0.8	22.66 ± 0.57 ^b^	8.12 ± 0.73 ^b,c^	1.39 ± 0.04 ^b^	3.82 ± 0.18 ^a^	64.00 ± 0.00 ^b^
	4.2	19.38 ± 0.02 ^a^	9.50 ± 1.66 ^c^	1.08 ± 0.02 ^a^	4.37 ± 0.16 ^b^	65.66 ± 1.53 ^b^
Foot	Control	22.68 ± 0.28 ^c^	12.85 ± 0.87 ^a^	1.71 ± 0.58 ^a^	2.42 ± 0.25 ^a^	60.33 ± 1.15 ^a,b^
	0.42	20.94 ± 0.25 ^b^	11.72 ± 0.52 ^a^	2.90 ± 0.06 ^b^	2.78 ± 0.14 ^a,b^	61.66 ± 0.58 ^b^
	0.8	21.18 ± 1.22 ^b^	14.07 ± 0.23 ^b^	1.56 ± 0.01 ^a^	3.51 ± 0.80 ^b^	59.66 ± 1.15 ^a^
	4.2	19.18 ± 0.49 ^a^	14.00 ± 0.36 ^b^	1.96 ± 0.01 ^a^	5.86 ± 0.51 ^c^	59.00 ± 0.00 ^a^
Gills	Control	32.15 ± 0.30 ^c^	4.50 ± 0.50 ^b^	1.96 ± 0.00 ^a^	2.39 ± 0.38 ^a^	59.00 ± 0.00 ^a^
	0.42	25.56 ± 0.40 ^b^	2.42 ± 0.40 ^a^	9.85 ± 0.00 ^d^	3.17 ± 0.36 ^a,b^	59.00 ± 0.00 ^a^
	0.8	25.42 ± 0.18 ^b^	4.36 ± 0.56 ^b^	7.43 ± 0.38 ^c^	3.80 ± 0.33 ^b^	59.00 ± 0.00 ^a^
	4.2	15.36 ± 0.06 ^a^	9.36 ± 0.50 ^c^	5.74 ± 0.19 ^b^	2.54 ± 0.54 ^a^	67.00 ± 0.00 ^b^
Intestine	Control	28.12 ± 0.59 ^c^	7.90 ± 0.57 ^a^	1.88 ± 0.21 ^a^	3.10 ± 0.55 ^a^	59.00 ± 0.00 ^a,b^
	0.42	27.91 ± 0.91 ^c^	6.72 ± 0.88 ^a^	2.41 ± 0.01 ^a^	6.29 ±2.28 ^a,b^	56.66 ± 4.04 ^a^
	0.8	21.11 ± 0.06 ^b^	9.58 ± 0.11 ^b^	3.62 ± 0.05 ^b^	5.03 ± 1.33 ^a,b^	60.66 ± 1.15 ^a,b^
	4.2	10.61 ± 0.22 ^a^	10.70 ± 0.71 ^b^	7.39 ± 1.12 ^c^	8.30 ± 2.28 ^b^	63.00 ± 1.00 ^b^
Mantle	Control	18.16 ± 0.24 ^a^	11.51 ± 0.94 ^b^	4.81 ± 0.69 ^a^	2.51 ± 0.50 ^a^	63.00 ± 0.00 ^b^
	0.42	17.66 ± 0.12 ^a^	11.33 ± 0.33 ^b^	9.22 ± 0.55 ^b^	2.79 ± 0.30 ^a^	59.00 ± 0.00 ^a^
	0.8	17.90 ± 0.42 ^a^	10.56 ± 0.30 ^b^	9.54 ± 0.58 ^b,c^	3.01 ± 0.67 ^a^	59.00 ± 0.00 ^a^
	4.2	18.86 ± 1.02 ^a^	8.30 ± 1.57 ^a^	10.31 ± 0.00 ^c^	3.53 ± 0.55 ^a^	59.00 ± 0.00 ^a^

Each value is the mean ± standard deviation of three replicates; values followed by the same superscript letter in the same column of same body part are not significantly different at *p* = 0.05.

## Data Availability

All data sets, on which the conclusions of the manuscript rely on, are present in the results section in the manuscript.

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
