# Peer review of "Tissue Distributions and Toxic Effects of Hexavalent Chromium in Laboratory-Exposed Periwinkle (Littorina littorea Linnaeus)"

_animals, 2023, doi:10.3390/ani13213412_

Round 1
Reviewer 1 Report
Comments and Suggestions for Authors
Review report on “Tissue distributions and toxic effects of hexavalent chromium in laboratory-exposed periwinkle (Littorina littorea Linaeus)”
A brief summary
This manuscript describes an experiment to measure the accumulation and effects on biochemical composition and histology of periwinkles exposed chronically to chromium.
General concept comments
Excellent introduction, although more precise information on the effects of chromium on the other organisms would be insightful. The concept of the investigation is important and covers a species that could be perhaps a useful biomonitor species, although this is not mentioned. The conclusions drawn let down the paper, as do the standard of the histology images, although it is an important topic to investigate.
Specific comments
Abstract
Line 15: Remove “although” from start of sentence.
Line 23 to 24: This is not accurate. The chromium concentration was not significantly different between the control and chromium-exposed animals, except in the foot tissue. Unless the results table 2 is incorrect with regard the indications of statistical significance of the subscript letters?
Introduction
Line 52: Perhaps rephrase or give more details on the ‘DNA tail’ and the assay that was carried out.
Lines 53 to 59: Please be more specific about the mutagenic effects in the three publications mentioned (16 to 19). If possible, mention the exposure concentrations.
Methods
Line 85-86: How was the background water concentration of chromium measured?
Line 89: Do you think the plastic containers would have adsorbed any chromium or leached any other contaminant into the experimental water?
Line 89-90: How did you choose these concentrations of chromium? Please state in the text. Please state in the text the salinity of the experimental water used. Which chromium salt was used for the exposures?
Line 94: Please state the software used for the Probit analysis.
Line 100: Were the plastic containers acid washed before treatment or treated in some way to remove chromium from previous experiments?
Line 103: Were the animals able to come out of the water? Did the frequency of them coming out of the water differ between treatment groups?
Line 110: Please use the numbering reference style.
Since the reader may not be familiar with periwinkle anatomy, please state reasons why the different tissues were selected and their importance to the animal, e.g. foot and mantle. Why did you not look at reproductive tissues?
Line 130 onwards: Fish tissues? Please correct in section title and throughout the section on proximate composition analysis.
Line 145: Font size
Line 154: typo: microtome.
Results
The data would be more appropriately shown as mean +/- SEM throughout.
The concentrations of chromium measured are very low indeed, compared with other studies. Are your units correct?
Line 192: It is not accurate to say the “the accumulation of total chromium was highest in the brain….etc”. The brain, gills, intestine, and mantle did not accumulate any chromium during the course of the laboratory exposure since the concentrations were not different from the control animals. Please rephrase this in the text.
Proximate compositions of the tissues section: Lines 200 to 240. This section would benefit from an overall statement concerning any trends seen throughout all of the tissues. For example, the percentage of protein was reduced with increasing chromium exposure concentration in all tissues, except for the mantle. And similarly for the other measurements.
On Plate 1. Brain. (rename to figure 2), please put the exposure concentrations on the figure in the corner of each photograph. Please put a scale bar in each photograph.
Plate 2. Gills. (rename to figure 3): how can you tell that the areas designated ‘areas of atrophy’ or ‘areas of necrosis’ are under atrophy or necrosis? Scale bars required. Did you photograph similar regions of the gill in each treatment group?
Plate 3. Mantle. (rename to figure 4): you have shown that in the control group, there is secretory cell degeneration. Please explain this in the text.
Plate 4. Digestive tissues. (rename to figure 5). Is this the intestine? Naming it digestive tissues needs clarification. In D2: how can you be sure that N is actually necrosis? The cells look no different from D1 except their arrangement is less convoluted.
Plate 5: Foot. (rename to Figure 6). The section taken for F3 contains broken tissue, and therefore any “degenerative areas” similar to those in F4 are not visible. Scale bars would be useful. The sections are difficult to compare as they look so different between the treatments – different scales or different parts of the tissues examined perhaps?
Discussion
Line 316 and 366: Please clarify “ differential accumulation”. Most of the tissues didn’t contain elevated chromium compared with controls.
Line 320 onwards. At the highest exposure concentration, the brain did not have a significantly higher concentration of chromium compared with the intestine or the muscle. The intestine did not have a significantly different concentration when compared with the mantle at any of the exposure concentrations. Therefore you can’t really say that the intestine accumulated chromium more than the mantle. Please change this in the text.
How do the chromium concentrations you measured compare with those of other studies, and with safety guidelines.
It would be useful for you to discuss why in most tissues, the control animals contained the same chromium as the exposed animals – could the exposed animals be excreting chromium? Are they accumulating the metal in a tissue not studied here? Is the chromium being adsorbed onto the shell? I find it surprising that the gills did not contain an elevated chromium level since they have high surface area exposed to the water.
Author Response
Response to Comments by Reviewer 1
A brief summary
This manuscript describes an experiment to measure the accumulation and effects on biochemical composition and histology of periwinkles exposed chronically to chromium.
General concept comments
Excellent introduction, although more precise information on the effects of chromium on the other organisms would be insightful. The concept of the investigation is important and covers a species that could be perhaps a useful biomonitor species, although this is not mentioned. The conclusions drawn let down the paper, as do the standard of the histology images, although it is an important topic to investigate.
Specific comments
ABSTRACT
Comment 1: Line 15: Remove “although” from start of sentence.
Response: Done
Comment 2: Line 23 to 24: This is not accurate. The chromium concentration was not significantly different between the control and chromium-exposed animals, except in the foot tissue. Unless the results table 2 is incorrect with regard the indications of statistical significance of the subscript letters?
Response: We agreed with the reviewer that this is not entirely correct. The tissue levels of Cr6+ differed significantly with respect to the tissue types while there was no significant effect of exposure concentration except for the foot tissue. This has been corrected in the revised manuscript. See lines 32-34 of the revised manuscript.
INTRODUCTION
Comment 3: Line 52: Perhaps rephrase or give more details on the ‘DNA tail’ and the assay that was carried out.
Response: Done. Please, see lines 63-66 of the revised manuscript.
Comment 4: Lines 53 to 59: Please be more specific about the mutagenic effects in the three publications mentioned (16 to 19). If possible, mention the exposure concentrations.
Response: The suggested details have been included in the revised manuscript. Please, see lines 63-82 of the revised manuscript.
METHODS
Comment 5: Line 85-86: How was the background water concentration of chromium measured?
Response: The levels of hexavalent chromium in the water samples collected from collection points were determined using an atomic absorption spectrophotometer. This detail has been included in the revised manuscript. See lines 107-109.
Comment 6: Line 89: Do you think the plastic containers would have adsorbed any chromium or leached any other contaminant into the experimental water?
Response: The plastic containers were acid-washed before use, and the control animals were also maintained in the same plastic containers, so we don’t expect the background levels of hexavalent chromium in the containers to have significant effects on the results.
Comment 7: Line 89-90: How did you choose these concentrations of chromium? Please state in the text. Please state in the text the salinity of the experimental water used. Which chromium salt was used for the exposures?
Response: The choice of the exposure concentrations was based on previous range-finding experiments performed in the laboratory. The test solutions were prepared from a 1000 mg/L stock solution of hexavalent chromium, which was previously prepared from a potassium dichromate salt. The details are now included in the revised manuscript. See lines 113-116 of the revised manuscript.
Comment 8: Line 94: Please state the software used for the Probit analysis.
Response: Probit analysis was performed using the Statistics Package for Social Sciences (SPSS), version 21. Please, see lines 121-123 of the revised manuscript.
Comment 9: Line 100: Were the plastic containers acid washed before treatment or treated in some way to remove chromium from previous experiments?
Response: The containers were acid-washed prior to use in experiments. This detail has been included in the revised manuscript. See lines 116-117.
Comment 10: Line 103: Were the animals able to come out of the water? Did the frequency of them coming out of the water differ between treatment groups?
Response: The containers were covered with muslin cloth to prevent the escape of animals but allow for access to air. This detail was included in the revised manuscript. See lines 130-132.
Comment 11: Line 110: Please use the numbering reference style.
Response: Done
Comment 12: Since the reader may not be familiar with periwinkle anatomy, please state reasons why the different tissues were selected and their importance to the animal, e.g. foot and mantle. Why did you not look at reproductive tissues?
Response: The selected tissues are the major organs of the animals, and possibly with the potential to bioaccumulate toxicants.
Comment 13: Line 130 onwards: Fish tissues? Please correct in section title and throughout the section on proximate composition analysis.
Response: We thank the reviewer for pointing our attention to this mistake. We actually worked with periwinkles. This has been corrected in the revised manuscript.
Comment 14: Line 145: Font size
Response: Corrected
Comment 15: Line 154: typo: microtome.
Response: it was a typo. The equipment used for the sectioning was a microtome. It has been written correctly in the revised manuscript. See line 185 of the revised manuscript.
RESULTS
Comment 16: The data would be more appropriately shown as mean +/- SEM throughout.
Response: We decided to report the data as ±SD considering that we had either three or four replicates. This was stated in the footnotes of each table or figure.
Comment 17: The concentrations of chromium measured are very low indeed, compared with other studies. Are your units correct?
Response: We decided to double-check and found out that the unit is actually µg/g and not µg/Kg
Comment 18: Line 192: It is not accurate to say the “the accumulation of total chromium was highest in the brain….etc”. The brain, gills, intestine, and mantle did not accumulate any chromium during the course of the laboratory exposure since the concentrations were not different from the control animals. Please rephrase this in the text.
Response: The sentence has been recasted.
Comment 19: Proximate compositions of the tissues section: Lines 200 to 240. This section would benefit from an overall statement concerning any trends seen throughout all of the tissues. For example, the percentage of protein was reduced with increasing chromium exposure concentration in all tissues, except for the mantle. And similarly for the other measurements.
Response: Done. See lines 234-239 of the revised manuscript.
Comment 20: On Plate 1. Brain. (rename to figure 2), please put the exposure concentrations on the figure in the corner of each photograph. Please put a scale bar in each photograph.
Response: The plates are now renamed as figures (Figures 2-6). The exposure concentrations and scaler bars have been inserted in the figures.
Comment 21: Plate 2. Gills. (rename to figure 3): how can you tell that the areas designated ‘areas of atrophy’ or ‘areas of necrosis’ are under atrophy or necrosis? Scale bars required. Did you photograph similar regions of the gill in each treatment group?
Response: The photographs were taken at the same region of the tissue.
Comment 21: Plate 3. Mantle. (rename to figure 4): you have shown that in the control group, there is secretory cell degeneration. Please explain this in the text.
Response: There was no observable lesion in the control animals and those exposed to 0.42 mg/L hexavalent chromium while there was degeneration of secretory cells and mantle epithelium in those exposed to 0.84 and 4.2 mg/L hexavalent chromium. This is well stated in the revised manuscript.
Comment 22: Plate 4. Digestive tissues. (rename to figure 5). Is this the intestine? Naming it digestive tissues needs clarification. In D2: how can you be sure that N is actually necrosis? The cells look no different from D1 except their arrangement is less convoluted.
Response: This has been renamed as intestine in the revised manuscript.
Comment 23: Plate 5: Foot. (rename to Figure 6). The section taken for F3 contains broken tissue, and therefore any “degenerative areas” similar to those in F4 are not visible. Scale bars would be useful. The sections are difficult to compare as they look so different between the treatments – different scales or different parts of the tissues examined perhaps?
Response: Now renamed as Figure 6 and scale bars inserted.
DISCUSSION
Comment 24: Line 316 and 366: Please clarify “differential accumulation”. Most of the tissues didn’t contain elevated chromium compared with controls.
Response: We meant differences in the levels of hexavalent chromium in the different tissue types. The sentences have been recast in the revised manuscript.
Comment 25: Line 320 onwards. At the highest exposure concentration, the brain did not have a significantly higher concentration of chromium compared with the intestine or the muscle. The intestine did not have a significantly different concentration when compared with the mantle at any of the exposure concentrations. Therefore you can’t really say that the intestine accumulated chromium more than the mantle. Please change this in the text.
Response: Indeed, there was no significant difference in the concentrations of hexavalent chromium in the brain, intestine, and mantle. The sentence has been recast in the revised manuscript.
Comment 26: How do the chromium concentrations you measured compare with those of other studies, and with safety guidelines.
Response: There have been wide disparities in the levels and patterns of tissue-specific accumulation of hexavalent chromium in aquatic organisms. Comparatively, the tissue levels of hexavalent chromium measured in this study were low. We have included the comparative data in other studies in the revised manuscript. See lines 427-434 of the revised manuscript.
Comment 27: It would be useful for you to discuss why in most tissues, the control animals contained the same chromium as the exposed animals – could the exposed animals be excreting chromium? Are they accumulating the metal in a tissue not studied here? Is the chromium being adsorbed onto the shell? I find it surprising that the gills did not contain an elevated chromium level since they have high surface area exposed to the water.
Response: As rightly mentioned, there was no significant difference in the levels of hexavalent chromium among the control and the exposed groups for most tissues. We stated in the manuscript that the exposure concentration did not affect tissue accumulation of hexavalent chromium in periwinkles. The mechanistic explanation for this observation was not included in the present study and would require further studies before a strong comment could be made on the observation.
Reviewer 2 Report
Comments and Suggestions for Authors
The present study investigated the effects of Cr6+ exposure on the tissue distribution, proximate composition, and histopathology of an aquatic mollusk, periwinkle (Littorina littorea). The animals were exposed to sublethal concentrations of Cr6+ (0.42, 0.84, and 4.2 mg/L) for 30 days after which the condition index, tissue accumulation, proximate composition, and histopathological effects were determined. The findings from the study indicate that Cr6+ is toxic to periwinkles. The potential for accumulation of Cr6+ in the major organs could therefore result in human exposure to Cr6+ through the consumption of chromium-contaminated periwinkles. The study is very interesting but it needs to be carefully revised before publication.
1. Line 22,37, 54, suggest authors delete the extra space.
2. Line 52, The presence of a tail of DNA usually indicates the DNA was damaged. Please clarify what the meaning of “increased DNA in the DNA tail”.
3. Line 85, If the background concentration of total chromium in water samples was tested from water samples in collection point? The authors should clarified that.
4. Line 88-90, Why the authors selected these concentrations of Cr6+? Reference the previous study or based on the pretest?
5. Line96-98, it is recommended to clarify the LC50 before the sub-lethal study (also in the results). Specify whether the three sublethal concentrations were of the 96 h LC50 of Cr6+?
6. In Materials and Methods: The preparation method of the exposed medium and the reagents used in the experiment should be provided.
7. In Materials and Methods:The true concentration of Cr6+ in the exposure medium should be measured and reflected in the article.
8. Line 125, “thirty-day exposure, twenty animals” should be “30-day exposure, 20 animals”.
9. Line 130,131, 135, what are fish tissues? If it should be “periwinkle tissues” or “soft tissues” tttt?
10. Lines 145-147: it is recommended to change the font format to be consistent with the previous text.
11. Lines 168: Suggest changing the title to mortality rate.
12. Figure1. Suggest delete the legend item and add the corresponding concentrations under the abscissa axis.
13. In table 2, It is usually distinguished by upper and lower case letters, not superscript or subscript.
14. Plate 1- Plate 5 should be Figure 2-6. The figures should be arranged in order, why the authors use Plate to instead of Figure? Figure title also should be revised and the resolution of these graphs need to be improved.
15. Lines 254.269.280.297.310: The font should be consistent with the before and after.
Comments on the Quality of English Language
No obvious errors.
Author Response
Response to Comments by Reviewer 2
The present study investigated the effects of Cr6+ exposure on the tissue distribution, proximate composition, and histopathology of an aquatic mollusk, periwinkle (Littorina littorea). The animals were exposed to sublethal concentrations of Cr6+ (0.42, 0.84, and 4.2 mg/L) for 30 days after which the condition index, tissue accumulation, proximate composition, and histopathological effects were determined. The findings from the study indicate that Cr6+ is toxic to periwinkles. The potential for accumulation of Cr6+ in the major organs could therefore result in human exposure to Cr6+ through the consumption of chromium-contaminated periwinkles. The study is very interesting but it needs to be carefully revised before publication.
- Line 22,37, 54, suggest authors delete the extra space.
Response: The extra spaces have been deleted in the revised manuscript
- Line 52, The presence of a tail of DNA usually indicates the DNA was damaged. Please clarify what the meaning of “increased DNA in the DNA tail”.
Response: The amount of DNA in the DNA tail is one of the parameters used in the comet assay to show genotoxicity. Other comet assay parameters that are routinely used include tail moment and tail length. We have made it clearer in the revised manuscript See line 63 of the revised manuscript.
- Line 85, If the background concentration of total chromium in water samples was tested from water samples in collection point? The authors should clarified that.
Response: Yes, the water samples were collected from the collection point. This information has been included in the revised manuscript. See lines 107-109.
- Line 88-90, Why the authors selected these concentrations of Cr6+? Reference the previous study or based on the pretest?
Response: The choice of the exposure concentrations was based on previous range-finding experiments performed in the laboratory. This information is now included in the revised manuscript. See lines 113-114.
- Lines 96-98, it is recommended to clarify the LC50 before the sub-lethal study (also in the results). Specify whether the three sublethal concentrations were of the 96 h LC50 of Cr6+?
Response: The sublethal concentrations were based on 96 h LC50. This was mentioned in the manuscript. See lines 125-126.
- In Materials and Methods: The preparation method of the exposed medium and the reagents used in the experiment should be provided.
Response: The test solutions were prepared from a 1000 mg/L stock solution of hexavalent chromium, which was previously prepared from a potassium dichromate salt. The details are now included in the revised manuscript. See lines 114-116 of the revised manuscript.
- In Materials and Methods: The true concentration of Cr6+in the exposure medium should be measured and reflected in the article.
Response: We really didn’t confirm the actual concentrations of Cr6+ in the exposure media. We feel very strongly that the stock solution was accurately prepared, and tried as much as possible to minimize pipetting errors during the addition of the stock solutions to the exposure media.
- Line 125, “thirty-day exposure, twenty animals” should be “30-day exposure, 20 animals”.
Response: Done. See line 153 of the revised manuscript.
- Line 130,131, 135, what are fish tissues? If it should be “periwinkle tissues” or “soft tissues” ?
Response: We thank the reviewer for pointing our attention to this mistake. We actually worked with periwinkles. This has been corrected in the revised manuscript.
- Lines 145-147: it is recommended to change the font format to be consistent with the previous text.
Response: Done.
- Lines 168: Suggest changing the title to mortality rate.
Response: Done. See line 199 of the revised manuscript.
- Figure 1. Suggest delete the legend item and add the corresponding concentrations under the abscissa axis.
Response: Done
- In table 2, It is usually distinguished by upper and lower case letters, not superscript or subscript.
Response: Done. The legend has been updated accordingly.
- Plate 1- Plate 5 should be Figure 2-6. The figures should be arranged in order, why the authors use Plate to instead of Figure? Figure title also should be revised and the resolution of these graphs need to be improved.
Response: Plates 1-5 are now numbered as Figures 2-6.
- Lines 254.269.280.297.310: The font should be consistent with the before and after.
Response: Done
Reviewer 3 Report
Comments and Suggestions for Authors
The manuscript titled "Tissue Distributions and Toxic Effects of Hexavalent Chromium in Laboratory-Exposed Periwinkle (Littorina littorea Linnaeus)" presents an interesting study on the effects of hexavalent chromium (Cr6+) exposure on the periwinkle, Littorina littorea. The research addresses an important environmental concern regarding the potential impacts of heavy metal pollution on aquatic organisms and human consumption of contaminated seafood. However, before this manuscript can be accepted for publication, there are several issues that need to be addressed:
Abstract needs to be revised, some pieces of information seem far-fetched. L22-24: “… The levels of Cr6+ in the tissues differed significantly both with preference to the exposure concentrations and the tissue types.” But some changes were observed only after the effect of the highest concentration of Cr. Summarisation of the abstract sounds “too pathetic” but couldn’t be derived from the manuscript.
Introduction: Please provide information on how much Cr can be met in the surface and marine waters and then explain why you chosed such a high concentration for the experiments.
You provided some information about the effects of Cr on different organisms, but there is a lack of information about mollusks. Please add it
Methods: Unfortunately, I couldn’t find the information on how you determined lipids.
Tables: Please check your results in Table1, three numbers are repeated in two different lines (17.5 and 21.0 mg/L)
The discussion needs to be deeply revised. As an example, consider the speculation about metallothioneins (L335-336). These proteins are not specific for Cr, moreover it can oppress their synthesis (Kimura, 2011).
Interpretation of Results: The manuscript should provide a more detailed interpretation of the results. For example, what are the potential ecological implications of the observed histopathological changes in periwinkles? How might these findings relate to broader environmental concerns?
Conclusions and Implications: The manuscript should conclude with a clear summary of the key findings and their broader implications for environmental science and human health.
Comments on the Quality of English LanguageMinor revision of English is required
Author Response
Response to Comments by Reviewer 3
The manuscript titled "Tissue Distributions and Toxic Effects of Hexavalent Chromium in Laboratory-Exposed Periwinkle (Littorina littorea Linnaeus)" presents an interesting study on the effects of hexavalent chromium (Cr6+) exposure on the periwinkle, Littorina littorea. The research addresses an important environmental concern regarding the potential impacts of heavy metal pollution on aquatic organisms and human consumption of contaminated seafood. However, before this manuscript can be accepted for publication, there are several issues that need to be addressed:
- Abstractneeds to be revised, some pieces of information seem far-fetched. L22-24: “… The levels of Cr6+ in the tissues differed significantly both with preference to the exposure concentrations and the tissue types.” But some changes were observed only after the effect of the highest concentration of Cr. Summarisation of the abstract sounds “too pathetic” but couldn’t be derived from the manuscript.
Response: The abstract has been rewritten in the revised manuscript.
- Introduction: Please provide information on how much Cr can be met in the surface and marine waters and then explain why you chose such a high concentration for the experiments.
Response: Provided in the revised manuscript. See lines 54-57.
- You provided some information about the effects of Cr on different organisms, but there is a lack of information about mollusks. Please add it.
Response: Now included in the revised manuscript. See lines 80-82.
- Methods: Unfortunately, I couldn’t find the information on how you determined lipids.
Response: Now included in the revised manuscript. See lines 173-176.
- Tables: Please check your results in Table1, three numbers are repeated in two different lines (17.5 and 21.0 mg/L).
Response: They are correct.
- The discussion needs to be deeply revised. As an example, consider the speculation about metallothioneins (L335-336). These proteins are not specific for Cr, moreover it can oppress their synthesis (Kimura, 2011).
Response: We have made efforts to revise some aspects of the discussion. However, regarding the roles metallothionein in detoxifying chromium toxicity in aquatic organisms, we feel that the proteins may still be involved in chromium detoxification in aquatic organisms since there are other studies that have reported increased metallothionein contents in fish exposed to hexavalent chromium, for example Velma and Tchounwou (2010), which has been cited in the revised manuscript.
- Interpretation of Results: The manuscript should provide a more detailed interpretation of the results. For example, what are the potential ecological implications of the observed histopathological changes in periwinkles? How might these findings relate to broader environmental concerns?
Response: Done in the revised manuscript.
- Conclusions and Implications: The manuscript should conclude with a clear summary of the key findings and their broader implications for environmental science and human health.
Response: The conclusion section has been revised accordingly.
Round 2
Reviewer 2 Report
Comments and Suggestions for Authors
The manuscript is improved and my comments adequately addressed. In my opinion, the article has met the requirements of the journal.
Author Response
No comment